# Prostaglandin E$_2$ dependent migration of human brain endothelial cells is mediated through Rho-Kinase-II

Gausal Azam Khan [1]*, Arjun Ghosh[2]

1 Department of Clinical Nutrition, College of Applied Medical Sciences, King Faisal University, Al Ahsa, Saudi Arabia, 2 Department of Biotechnology, Brainware University, Kolkata, India

* gkhan@kfu.edu.sa, gausalk@gmail.com

## Abstract

Prostaglandin E$_2$ (PGE$_2$), that plays a crucial role in angiogenesis as well as in ischemic and inflammatory disorders of the brain, is associated with breakdown of the blood-brain barrier (BBB). Previously, we had shown that PGE$_2$-induced human brain endothelial cells (HBECs) migration, and works in a cooperative manner through its three receptors (EP$_2$, EP$_3$ & EP$_4$). However, the detailed signaling mechanism of PGE$_2$-induced HBECs migration remains obscure. In this present study, we investigated the signaling pathway of actin dynamics/polymerization and migration of HBECs by PGE$_2$ in vitro. Expression of ROCK was analyzed by ELISA and RT-PCR. Actin polymerization was evaluated by NBD-phallacidin immunofluorescence staining. HBECs expressed only ROCK II. PGE$_2$ (100 pM) induced ROCK II expression occurs in dose-and-time-dependent manner. ROCK II inhibition by Y27632 (150nM), as well as ROCK II silencing significantly attenuated PGE$_2$-induced migration of HBECs. We further showed that pretreatment of PKA inhibitor (H-89; 0.5 μM) or adenylate cyclase inhibitor (ddA; 1μM) completely inhibited PGE$_2$-induced ROCK II activity. Furthermore, PGE$_2$-induced MLC phosphorylation also occurs in a time-dependent manner. However, pretreatment of ROCK II inhibitor or silencing of ROCK II significantly abrogated PGE$_2$-induced MLC phosphorylation as well as F-actin polymerization. Our ex-vivo aortic ring angiogenesis study also showed that pretreatment of ROCK II inhibitor significantly inhibited ECs sprouting. These results suggest that PGE$_2$-induced HBECs migration is mediated through PKA, ROCK II and MLC phosphorylation as well as F-actin polymerization, indicating that modulation of these pathways may aid in the future treatment of dysregulated angiogenesis in cerebrovascular diseases.

## Introduction

Angiogenesis is a well-orchestrated physiological phenomenon where the formation of new capillaries arises from pre-existing blood vessels of mostly in ischemic/

**Data availability statement:** All relevant data are within the manuscript and its Supporting Information files.

**Funding:** This work was supported by the Deanship of Scientific Research, Vice Presidency for Graduate Studies and Scientific Research, King Faisal University, Saudi Arabia [Grant No. KFU 252130 to GAK].

**Competing interests:** The authors have declared that no competing interests exist.

**Abbreviations:** $PGE_2$, Prostaglandin $E_2$; HBECs, Human brain endothelial cells; ROCK-II, Rho-Kinase-II; PKA, Protein Kinase A; WB, Western blot; BBB - Blood-brain barrier; ddA 2', 2'-dideoxyadenosine; PGES, Prostaglandin synthases; PVDF, Polyvinylidene fluoride; HGF, Hepatocytes growth factor; IBMX, 3-isobutyl-1-methyl-xanthine; PBS, Phosphate buffered saline; PMSF, Phenylmethylsulphonylfluoride; BSA, Bovine serum albumin; MLC, Myosin light chain; COX, Cyclooxygenase; FBS, Fetal bovine serum; cAMP, Cyclic adenosine monophosphate; NaF, Sodium fluoride; $Na_3VO_4$, Sodium vanadate; ECL, Enhanced chemiluminescence; DMEM, Dulbecco's Modified Eagle's Medium.

hypoxic tissues [1]. In ischemic hearts or limbs, therapeutic angiogenesis promotes blood vessels formation. Angiogenesis is initiated by activation of endothelial cells (ECs), followed by ECs migration and proliferation [2].

ECs migration is a complex and highly organized process whereby some ECs [3] abandon the vascular bed in response to specific extracellular stimuli. These migratory ECs then infiltrate the basement membrane and reach the interstitial space where they proceed to differentiate and rearrange themselves to produce mature blood vessels [4,5]. The ability of ECs to readily migrate in response to external stimuli is a prerequisite event for successful angiogenesis during physiological as well as pathological conditions [6]. Therefore, the elucidation of the intracellular events that regulate the migratory response of ECs in response to extracellular signals is critical. We wish to modulate the angiogenic response for therapeutic purposes.

Prostaglandin $E_2$ ($PGE_2$) is a lipid signaling molecule produced by the enzymatic oxidation of arachidonic acid by the cyclooxygenase (COX) and $PGE_2$ synthase pathways [7]. Once generated, $PGE_2$ influences cellular functions via the activation of distinct signaling pathways which are triggered by its binding to four subtypes of G-protein coupled receptors (GPCRs) $EP_1$, $EP_2$, $EP_3$ and $EP_4$ [8]. Despite the evidence implicating overexpression of COX-2 and mPGES-1 in tumor-and inflammation-associated angiogenesis, the signaling pathways involved in the angiogenic responses of ECs to $PGE_2$ are still not well understood.

Small GTPase binding proteins of the Rho family have been implicated in the regulation of cell motility in a variety of cell types. Reduction in mechanosensitive ion channel transient receptor potential vanilloid (TRPV4) level in EC activates Rho/ROCK signaling, and results in enhanced proliferation, migration and aberrant tube formation [9]. ECs migration requires the contraction of actiin-myosin fiber which results in the rearrangement of stress fibers and the formation of focal adhesions, all of which are regulated by ROCK [10]. Activation of ROCK enhances actin-myosin contraction in ECs by promoting the phosphorylation of MLC [11]. Growth factor-dependent stimulation of RhoA activity results in activation of Rho-Kinase followed by polymerization and contraction of actin-myosin fibers [12]. It has been reported that the inhibition of ERK signaling, reduces tube formation of ECs and thereby suppresses angiogenesis through the ROCK-MLC pathway [13].

Previously, we have reported that $PGE_2$-induced migration of HBECs was mediated through the cooperation of its three major receptors, $EP_2$, $EP_3$ and $EP_4$, however, the exact signaling mechanism of $PGE_2$-induced migration of HBECs has not yet been explored.

Therefore, we hypothesized that $PGE_2$-induced migration of HBECs is mediated through Protein Kinase A- ROCK II- myosin light chain phosphorylation (MLC).

This study assessed whether PGE2-induced migration is mediated through Rho kinase, specifically ROCK II or not, which phosphorylates MLC and facilitates the migration of HBECs.

## Materials and methods

### Animals and ethical considerations

All the experiment cited in this manuscript titled "Prostaglandin $E_2$ dependent migration of Human Brain Endothelial Cells mediated through Rho-Kinase-II" received ethical clearance from the Internal Review Board at King Faisal University, Al Ahsa, SA (KFU-REC-2024-MAY-ETHICS238). The experiments involved the use of Swiss Albino mice (weighing ~25-30g), which were handled under the careful oversight of a licensed veterinarian, ensuring compliance with the established protocol. All procedures adhered to the principles outlined in the Helsinki Declaration of 1964, with no modifications allowed during the study.

The mice were housed in the animal facility of the institute and provided with chow diet and water *ad libitum*. All animal procedures were performed under urethane (1.2g/kg) induced general anesthesia to minimize suffering as described previously by us [14].

### Reagents

Endothelial cell medium with culture boost (cat# 4Z0-500R), serum-free medium kit without growth factors (cat# SF-4Z0-500S), and attachment factor (cat# 4Z0-210) were obtained from Cell Systems (Kirkland, WA, USA). 16,16-dimethyl $PGE_2$ (cat# 14750), protein kinase A inhibitor amide (PKAI; cat# P6062), 3-isobutyl-1-methyl-xanthine (IBMX; cat# 17859), ATP (cat# A2383), β-actin antibodies (cat# A2228) and Enhanced avian reverse transcriptase [eAMV™ RT] (cat# A4464-1KU) were obtained from Sigma Chemical Co (St. Louis, MO, USA). Hepatocytes growth factor (HGF; cat# 249-HG) was purchased from R&D System (Minneapolis, MN, USA). ROCK II inhibitor Y27632 (cat# 1254) was obtained from Tocris Corp (Minneapolis, MN, USA), H89 (cat# 371963), forskolin (cat# 344282), and 2,5´- dideoxyadenosine (ddA; cat# 288104) were obtained from Calbiochem (La Jolla, CA, USA). $^{32}$P-adenosine orthophosphate was purchased from Amersham Life Science (Arlington Heights, IL, USA). Transwell chemotactic chambers were purchased from Costar (Corning, NY, USA; cat# CLS3422). The oligofectamine/plus transfection reagent (cat# 12252011), Tris-Glycine precast gels, and SeeBlue2 prestained protein marker (cat# LC5925) were obtained from Invitrogen (Carlsbad, CA, USA). X-ray films were purchased from Phoenix Research Products (Candler, NC, USA; FBX57). Anti-ROCK II antibody (D-11) was purchased from Santa Cruz Biotechnology (cat# SC-398519; Santa Cruz, CA, USA). Anti-phospho myosin light chain (Ser19) was obtained from Cell Signaling Technology (cat# 3671; MA, USA), anti-myosin light chain 2 antibody was purchased from Abcam (cat# ab79935; Abcam, Cambridge, MA, USA). ROCK activity assay Kit (cat# ab211175) was purchased from Abcam, Cambridge, MA, USA. Invitrogen ™Molecular Probe™ NBD Phallacidin was obtained from Fisher Scientific, USA (cat# Invitrogen N354). The enhanced chemiluminesence system (cat# SC2048) was purchased from Santa Cruz Biotechnology (Santa Cruz, CA, USA). Diff-Quik staining kit (cat# CA53000−52) was purchased from VWR (Randor, PA, USA). RNeasy mini kit (cat# 74104) was purchased from Qiagen (Valencia, CA, USA).

### Endothelial cell culture

Primary human brain endothelial cells (HBECs) were purchased from cell systems. Cells were grown in attachment-factor coated tissue culture flasks and incubated in an atmosphere of 5% $CO_2$ and 37°C in endothelial tissue culture medium containing 10% fetal bovine serum (FBS). Cells were used between passages 2–10.

### Migration assay

HBECs migration was assayed as described previously by us using Costar transwells (6.5 mm diameter; 8-µm pore size) [7]. Briefly, semi-confluent HBECs were washed, trypsinized to obtain single-cell suspension and suspended in M199 media containing 0.1% bovine serum albumin (BSA). Unless otherwise stated, 1x105 cells were poured in fibronectin coated filters and placed in upper compartment of the transwell chamber and the lower chambers were filled with 0.3 ml

of serum-free M199 media containing indicated amounts PGE$_2$ (100 pM), HGF (1 ng/ml). In selected experiments HGF (1 ng/ml) was used as a positive control. Cells were allowed to migrate in a humidified incubator with 5% CO$_2$ at 37°C for indicated time periods. Filters were removed; upper side was wiped off with cotton applicator to remove nonmigrated cells. Migrated cells were fixed, and stained with Diff-Quik and counting of 5 randomly chosen fields (40x) was done for quantification.

## Transient RNA interference (RNAi) transfection

SMART pool RNAi from Dharmacon RNA Technologies (Lafayette, CO, USA) targeting ROCK II and SMART pool non-specific RNAi were used as control in this experiment. Briefly, HBECs (4 10$^4$/well) were seeded 24h prior to transfection. Cells were transfected using the Oligofectamine Plus reagent system (32μl:12μl) with 2 nmol of SMART pool RNAi in a final volume of 2 ml OPTI-MEM under serum-free condition and incubated at 37°C under 5% CO$_2$ in an incubator. After 24h of transfection cells were used for migration assay.

## Total RNA isolation

Total RNA was isolated by RNeasy Mini Kit (Qiagen Inc., Valencia, CA, USA) according to the manufacturer's protocol. Total RNA was isolated was reverse transcribed using an enhanced AMV reverse transcriptase system, according to the manufacturer's instructions. First-strand cDNA synthesis and PCR were performed as described previously [15].

ROCK I was amplified using specific 24-mer forward and reverse primers (5'-TGGTGAAACACCAGAAGGAGCTGA-3' and 5'-TAGCGCCCTGCAAAGTTTAT-3') respectively). Similarly, ROCK II was amplified using 20-mer forward and reverse primers (5'-TAGCGCCCTGCAAAGTTTAT-3' and 5'-CCGAATGGACTGGTTTTGTT-3') respectively. The conditions were: initial denaturation at 94°C for 5 min, followed by 35 cycles of 94°C for 35s (denaturation), 57.3°C for 35s (annealing), and 72°C for 50s (extension); and final elongation at 72°C for 5 min. In addition, the 18S rRNA gene (GenBank accession no.: X01117) was amplified as a reference by using specific 18-mer forward and reverse primers (5'-GCCCGAGCCGCCTGGATA-3' and 5'-CCGCCGCATCGCCAGTC-3') respectively under the following conditions: initial denaturation at 94°C for 5 min and 33 cycles of denaturation at 94°C for 35s, annealing at 55°C for 35s, and extension at 72°C for 50s. The PCR products were electrophoresed on 1% agarose gel and were visualized by staining with ethidium bromide [7].

## Enzyme linked immunosorbent assay (ELISA)

After stimulation with PGE2, cells were treated with ROCK II inhibitors—Y27632 and cells were then scraped off in ice-cold phosphate-buffered saline containing 5 nM NaF and Na$_3$VO$_4$ and centrifuged at 1500 rpm at 4 °C. Collected cells were lysed in RIPA buffer (50 mM Tris-HCl, pH 7.4 containing 1% NP-40, 0.25% sodium deoxycholate, 150 mM NaCl, 1 mM EDTA, 1 mM NaF) containing 10 μg/ml aprotinin and leupeptin respectively plus 1 mM PMSF. Following lysis, protein concentrations of the lysates were quantified by Bradford method using BSA as standard. ELISA was performed using HBECs and SMC cell lysate as described previously [16]. Briefly 10 μg of cell extract was coated onto the assay plate using 0.5 M carbonate buffer (pH 9.6) and incubated at 37°C for an hour. After incubation with specific primary and HRP-conjugated secondary antibodies, 1 mg/ml *p*-nitrophenyl phosphate was used as a substrate for colour development and absorbance measured at 450 nm. Optical density (OD) values were used to analyse the relative expression of the specific proteins between samples.

## Immunoblotting

The cells were then scraped off in ice-cold phosphate-buffered saline containing 5 nM NaF and Na$_3$VO$_4$ and centrifuged at 1500 rpm at 4 °C. Collected cells were lysed in RIPA buffer (50 mM Tris-HCl, pH 7.4 containing 1% NP-40, 0.25% Sodium

deoxycholate, 150 mM NaCl, 1mM EDTA, 1 mM NaF) containing 10 μg/ml aprotinin and leupeptin respectively plus 1mM PMSF. Following lysis, protein concentrations of the lysates were quantified by Bradford method using BSA as standard. Equal amounts of proteins were loaded on to two separate 12% precast ges and electrophoresis were performed. Following electrophoresis proteins were transferred onto PVDF membranes & immunoblotted with anti-MLC, anti-phospho MLC antibodies respectively, Membranes were washed and probed with anti-rabbit HRP secondary antibody (1:5000). Immunoreactive proteins were visualized by enhanced chemiluminescence (ECL) detection system. β-actin antibody was used as loading control.

### ROCK II activity assay

Briefly, $PGE_2$-induced HBECs were incubated in the presence or absence of different inhibitors or activators (ddA, H89, forskolin and HGF) in MYPT1-coated wells for 60 min at 30°C. Anti-phospho-MYPT1 (Thr696) antibody was added and incubated for 1 hr at RT. HRP-conjugated secondary antibody was added and incubated for 1 hr at RT, followed by washing and addition of substrate solution and incubated for further 20 min followed by addition of stop solution Immediately after the reaction, the absorbance was measured at OD 450 nm.

### Immunocytochemistry

HBECs grown in coverslips were stimulated with $PGE_2$ or vehicle for 15 minutes. Cells were washed, fixed with 4% paraformaldehyde and permeabilized with 0.1% Triton-X 100 for 5 min. Cells were incubated in PBS containing 0.5% BSA and 0.1% Triton-X 100 for 30 min. NBD-phallacidin (1:100) (Molecular Probe) was added and incubated for 30 min at RT. Monolayers were extensively washed with PBS, dried, mounted and examined under fluorescence microscopy.

### Aortic ring assay

To assess the role of $PGE_2$ and ROCK II in angiogenesis, we have employed the aortic ring assay as previously described [17]. This study was done on adult Swiss Albino mice weighing 25-30g.The mice were housed in the animal facility of the institute and provided with chow diet and water *ad libitum*. Mice aorta was isolated from the diaphragm and cut into 1mm long rings and washed with ice-cold PBS. Mice aortic rings were incubated with $PGE_2$ and ROCK II inhibitor Y27632. 150 μl of growth-factor reduced matrigel was added in 24 well plates, followed by placement of aortic rings and covered with another set 150 μl matrigel to cover the aortic rings and incubated in Dulbecco's Modified Eagle's Medium (DMEM) with 10% FBS per well and incubated at 37°C in a humidified incubator for 7 days. Branch of vasculature or cell sprouting were observed under an inverted microscope [17].

To quantify angiogenic sprouting, we measured the radial distance of vessel outgrowth using ImageJ software as described previously [18,19]. Phase contrast images of the aortic rings were first processed by subtracting the background using a rolling ball radius of 700 pixels to enhance contrast and reduce noise. The threshold was then adjusted to highlight only the sprouts, excluding the aortic ring and background artifacts. After setting the image scale based on the microscope's resolution (0.645 μm/pixel), a circle was drawn around the outer edge of the sprouting vessels and another around the original aortic ring. The radial distance of the sprouts was determined by subtracting the radius of the inner circle (aortic ring) from that of the outer circle (vessel outgrowth), providing a reproducible metric of angiogenic response under each treatment condition.

### Statistical analysis

All experiments were repeated at least five times unless otherwise stated. Data are presented as means ± SEM. The statistical significance of differences between experimental groups was determined by performing one-way ANOVA followed by Bonferroni's multiple compression tests; $p < 0.05$ was considered statistically significant.

## Results

### Rho-associated protein kinase (ROCK) expression profile in HBECs

ROCK is a key regulator of actin organization and cellular migration in most cells. Therefore, an attempt was made to investigate the expression profile of ROCK subtypes in HBECs in both post transcription and post translation level by RT-PCR and ELISA respectively. Smooth muscle cells (SMC) were used as positive control. Fig 1a. ROCK II mRNA is expressed in both HBECs and SMC,whereas ROCK I mRNA is only expressed in SMC. The 18S mRNA was used as the loading control. To determine whether similar findings were observed at the post-translational level, ELISA was performed using both ROCK I and ROCK II antibodies for expression analysis. Fig 1b. Our data showed that ROCK II was detected in both HBECs and SMC. Fig 1c. whereas no expression of ROCK I was observed in HBECs. As such, we focused ROCK II on PGE$_2$-induced migration of HBECs for the rest of the experiments described below unless otherwise stated.

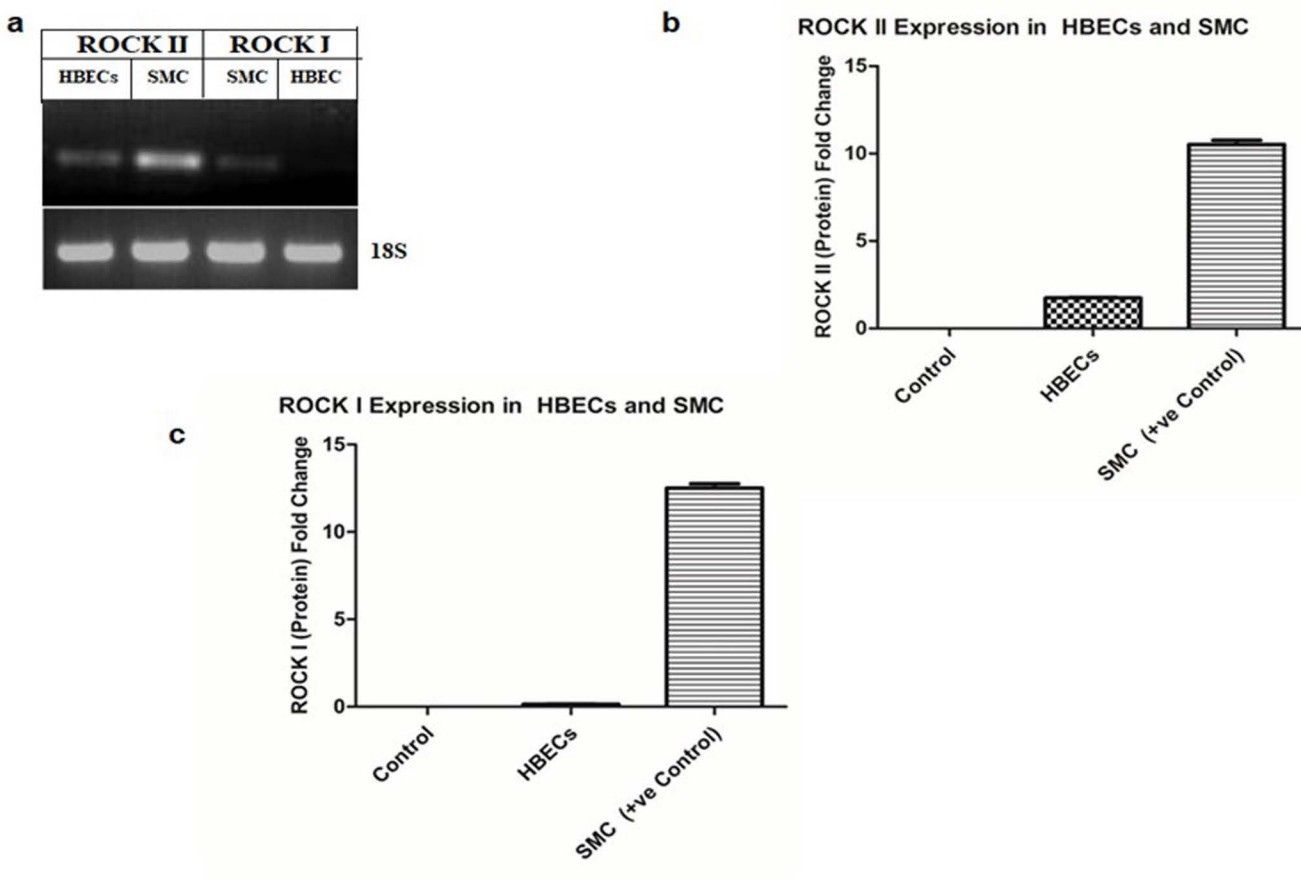

**Fig 1. Differential expression of ROCK I and ROCK II in HBECs and SMC. (a)** Total RNAs were isolated from the HBECs and SMC. ROCK I & II were amplified using specific 24-mer forward and reverse primers respectively, electrophoresed through agarose gel and visualized by ethidium bromide. Equal amounts of loading were confirmed by comparing the 18S fragment of mRNA. **(b-c)** HBECs and SMC cells were lysed, and equal amounts (10 μg) of extracts were coated onto plate followed by ELISA with specific antibodies of ROCK I & **II.** Triplicate experiments were performed, and the results are from an average titter of the respective antibody.

## PGE$_2$-induced ROCK II expression

Previously, we had reported that PGE$_2$ facilitates the migration HBECs in a dose-and-time dependent manner to exert its chemotactic effect [7]. HBECs showed maximum migratory response by PGE$_2$ at 100nM. To elucidate whether the PGE$_2$ exerts similar response in downstream signaling molecules or not, experiments were carried out in different dosage of PGE$_2$ first. Fig 2a. ROCK II expression was maximum in response at 0.1nM PGE$_2$ then gradually decreased and attained normalcy at 10nM.

An experiment was also conducted to determine the time-dependent effect of PGE$_2$ on ROCK II expression. It was found that the expression of ROCK II in HBECs in response to 0.1nM PGE$_2$ was maximum. Fig 2b. Also it was found that the effect of PGE$_2$ on ROCK II was maximum in 15 minutes and the effect was sustained till 30min, then diminished when compared to control. As such, we used 15-min time point for all subsequent studies unless otherwise stated.

## PGE$_2$-induced migration of HBECs is mediated through ROCK II

To evaluate the functional significance of PGE$_2$-induced migration of HBECs through ROCK II, we employed the RNAi technology to silence the expression of ROCK II and determine its contribution to the pro-migratory effects of PGE$_2$. The HBECs were transiently transfected with a pool of siRNA duplexes that specifically targeted the ROCK II (ROCK II-siRNA) or with non- specific siRNA (NS-siRNA) and the subsequent migratory response to PGE$_2$ was assayed.

Fig 3a. HBECs transfected with ROCK II-siRNA showed significant inhibition of migration in response to PGE$_2$ when compared to the control. The HGF was used as a positive control [7,20]. In contrast, NS-siRNA had no effect. Control studies with HGF did not show any a ppreciable change in the migratory response.

To validate the above results, we employed the pharmacological approach where we've pre-incubated (15min) HBECs with selective ROCK II inhibitor (Y27632; dose 150nM) and the pro-migratory effects of PGE$_2$ was assayed. Fig 3b. Our results showed that exposure of HBECs to 150nM Y27632 (selective ROCK II antagonist) significantly inhibited PGE$_2$ induced migration when compared to the control ($p < 0.50$). Fig 3b. In comparison, HGF-induced migration was not

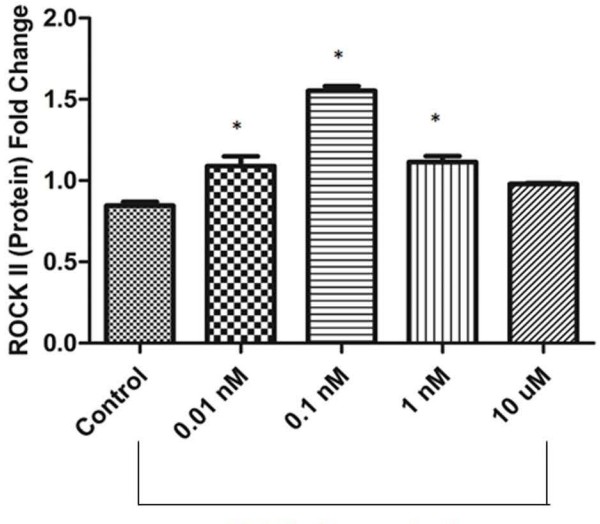
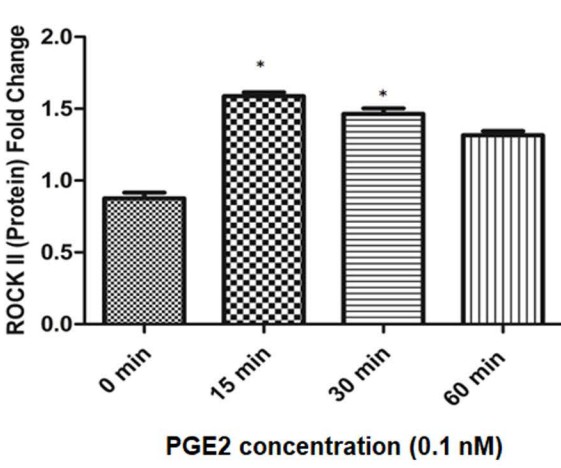

**Fig 2. PGE$_2$-induced stimulation of ROCK II in dose-and-time dependent manner. (a)** HBECs were treated with varying concentrations of PGE$_2$. Cells were lysed and equal amounts of proteins (10 µg) were coated followed by ELISA. **(b)** HBECs were treated with fixed concentration of PGE$_2$ (0.1nM) for the indicated time point and ELISA was performed with ROCK II antibody as previously described.

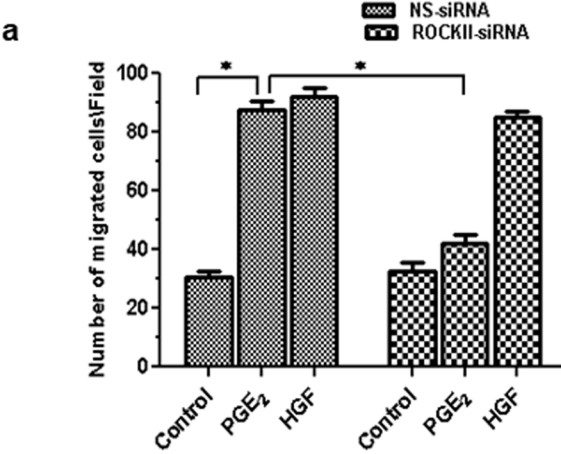
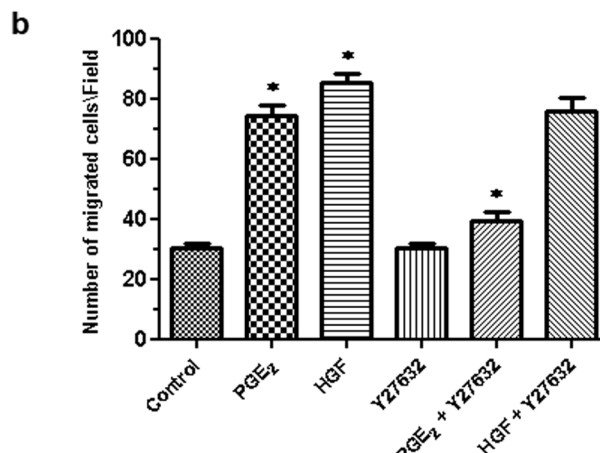

**Fig 3. PGE₂-induced migration of HBECs is mediated through ROCK II. (a)** Post-transient transfection with specific siRNA for ROCK II or non-specific RNA, migration of HBECs in response to different agonists, i.e., PGE₂ and HGF was monitored. After removal of the non-migrated cells from the filter, migrated cells were quantified by counting stained nuclei in 5 random fields (40x)/filter. Results are presented as mean ±SEM (n = 5/group/condition) from 5 independent experiments. **(b)** HBECs were transfected with specific ROCK II siRNA and/or scrambled (non-specific) siRNA followed by treatment with PGE₂. As described before, cells were lysed, and lysed cells was used as antigen to perform ELISA. Immunoreactive protein bands were visualized by the ECL reagent system. Results are from representative experiment from 5 independent experiments. **(c)** Migration of HBECs were analyzed and monitored for 8h with/without the presence of specific inhibitor of ROCK II (Y27632) followed by treatment with PGE₂ and HGF. After removal of the non-migrated cells from the filter, migrated cells were migrated cells were quantified by counting stained nuclei in 5 random fields (40x)/filter. Results are presented as mean ±SEM (n = 5/group/condition) from 5 independent experiments. One-way ANOVA revealed statistically significant differences in results (*P ‹ 0.05) vs. basal migration.

affected by pre-incubation with ROCK II inhibitor. These results suggest that PGE₂-induced migration of HBECs is mediated via ROCK II activation.

## PGE₂-induced ROCK II activation is mediated via cAMP-PKA

Previously, we showed the involvement of cAMP & PKA in PGE₂-induced migration of HBECs [7]. Hence, we explored whether cAMP-PKA axis had any role in PGE₂-induced ROCK II activation and subsequent migration of HBECs or not. Fig 4a. Our results showed that pretreatment of cAMP inhibitor ddA (1μM) for 30 min prior to exposure to PGE₂ significantly diminished PGE₂-induced ROCK II activation as compared to the control (p < 0.05). Forskolin, a direct activator of adenylate cyclase, was used as a positive control.

It is an established fact that cAMP activates PKA. Therefore, an attempt was made to establish whether cAMP dependent activation of PKA activated ROCK II or not

Using a pharmacological approach, we pretreated the HBECs with H-89 (a selective inhibitor of PKA; 0.5 μM) for 30 min and ROCK II activity was assayed. Fig 4b. Our data showed that pretreatment of H-89 significantly inhibited ROCK II activity when compared to the control (p < 0.05). Fig 4b. In contrast, HGF-induced ROCK II activity was not altered by pre-treatment with PKA inhibitor. These results suggest that PGE₂-induced ROCK II activation is mediated through cAMP-PKA axis.

## Effects of PGE₂ on activation of myosin light chain and polymerization of actin filament

Migration of cells is fundamental process for successful angiogenesis. To initiate the migratory response in a cell, the formation of a membrane protrusion (formation of lamellipodia and filopodia) at the front edge the migratory cells, is essential

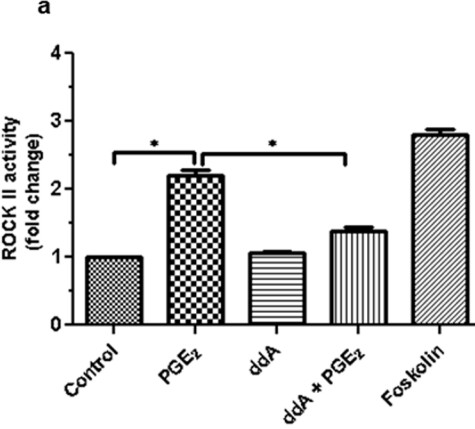
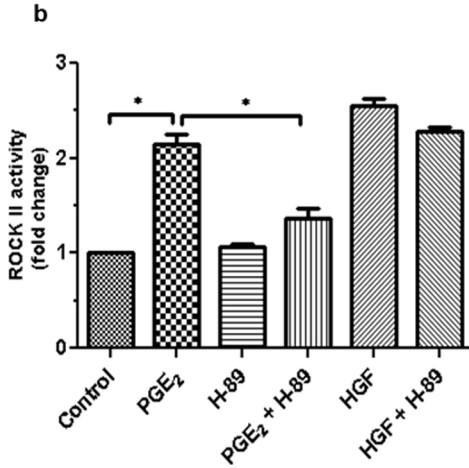

**Fig 4. PGE2-induced ROCK II activity is mediated through protein kinase A. (a)** HBECs were treated with cAMP inhibitor (ddA) in the presence or absence of PGE2 and ROCK II activity was measured by the enzyme-based immunoassay (colorimetric) as per the protocol supplied by the vendor and fold change is compared between the various treatment groups. Foskolin was used as positive control. **(b)** Similarly, HBECs were incubated with PGE2 and HGF in presence or absence of PKA inhibitor (H89) and Rho Kinase II activity was measured. Results are presented as mean ±SEM (n = 5/group/condition) from experiments with each condition being tested in quintuplet and are pooled from 5 independent experiments. One-way ANOVA revealed statistically significant differences in results (*P ‹ 0.05).

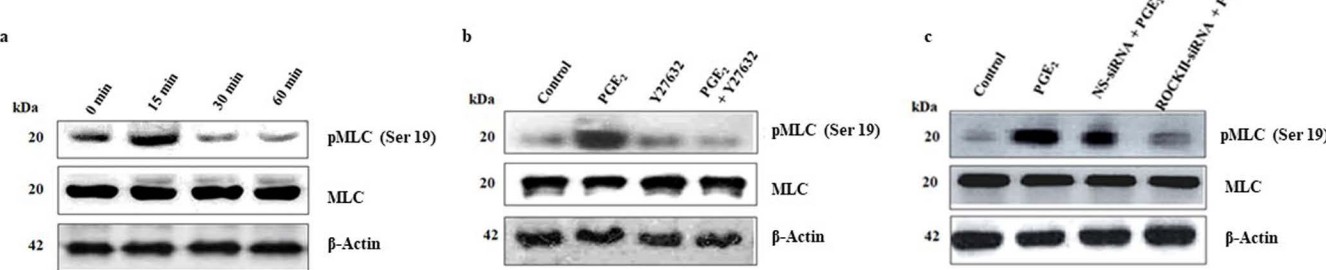

**Fig 5. PGE2-induced ROCK II activity via activation of myosin light chain. (a)** HBECs were treated with PGE2 for varying time periods, and equal amounts of lysates (10 µg) were resolved by PAGE and transferred onto PVDF membrane and immunoblotted with specific phospho-MLC (ser19) and total MLC antibody separately in two gels. Both the blot (s) were stripped and reprobed with β-actin antibody to verify equal loading of protein/lane. Immunoreactive protein bands were visualized by the ECL reagent system. Results are representative from 5 independent experiments. **(b)** HBECs were treated with PGE2 in the presence and absence of specific ROCK II inhibitor Y27632, equal amounts of cell lysates (10 µg) were resolved by PAGE, transferred onto PVDF membrane and immunoblotted with specific phospho-MLC (ser19) and total MLC antibody. Blot (s) were stripped in stripping buffer and WB for total amount of MLC and β-actin were done to verify equal amounts of loading of proteins/lane. Results are representative of 5 independent but separate experiments. **(c)** HBECs were transiently transfected with specific ROCK II siRNA and/or scrambled (non-specific) siRNA followed by treatment with PGE2. Cell lysates were electrophoresed, transferred onto PVDF membrane and immunoblotted with phospho-MLC (ser19) and MLC antibodies. Blot (s) were stripped and reprobed with β-actin antibody to verify equal loading of protein/lane. Immunoreactive protein bands were visualized by the ECL reagent system. Results are representative of 5 independent experiments.

and which is initiated by the phosphorylation of the regulatory light chains of myosin II as well as actin polymerization. Therefore, we explored whether PGE2-induced MLC phosphorylation was mediated through ROCK II. Firstly, experiments were conducted to determine the time-dependent effect of PGE2 on MLC phosphorylation. Fig 5a.Tthe phosphorylation of MLC (pMLC) in response to PGE2 (100 pM) was in a time-dependent manner as compared to control. Total MLC protein and β-actin was used as the loading control.

Therefore, we have adopted both the pharmacological as well as genetic approach to determine the role of ROCK II in PGE$_2$-induced pMLC in HBECs. In the pharmacological approach, we pretreated HBECs with ROCK II specific inhibitor (Y27632) and studied the PGE$_2$-induced pMLC. Fig 5b. Our data showed that pretreatment of Y27632 for 30 min significantly abrogated PGE$_2$-induced pMLC when compared to the control. For the genetic approach, we suppressed the expression of ROCK II by using siRNA (details discussed in method section) and then studied the PGE$_2$-induced MLC phosphorylation. Fig 5c. Our results showed that silencing of ROCK II significantly abrogated MLC phosphorylation in HBECs. Fig 5c. In contrast, NS-siRNA had no effect. Total MLC protein and β-actin were used as the loading control (s) in all the above-mentioned experiments. These results suggest that PGE$_2$-induced MLC phosphorylation is mediated through ROCK II.

Migration of cells is mostly driven by the polymerization of actin filaments (F-actin). Therefore, we investigated the PGE$_2$-induced F-actin polymerization in HBECs by both pharmacological as well as genetic approach. F-actin polymerization was studied by immunofluorescence method using NBD- phallacidin conjugate which specifically binds with F-actin filaments of cells (detailed in methods section). Fig 6a. Our data shows that pretreatment with Y27632 (ROCK II selective inhibitor) as well as H-89 (PKA inhibitor) significantly inhibited PGE$_2$-induced F-actin polymerization when compared to the control. For the genetic approach, we suppressed ROCK II by using siRNA (details discussed in method section) and then studied the PGE$_2$-induced F-actin polymerization. Fig 6b. Our results showed that silencing of ROCK II significantly abrogated F-actin polymerization in HBECs. Fig 6b. In contrast, NS-siRNA had no effect. These results suggest that PGE$_2$-induced F-actin polymerization is also mediated through PKA-ROCK II pathway.

## PGE$_2$-induces angiogenic sprouts through ROCK II

We used ex-vivo mouse aortic ring sprouting assay (detailed in methods section) to prove the proangiogenic effects of PGE$_2$. Fig 7. Our results showed that treatment of PGE$_2$ (100 pM) for 7 days promoted sprouting from the aortic ring as evidenced by the increased area of vessel outgrowth when compared with control. Fig 7. However, pretreatment of Y27632 (specific ROCK II inhibitor) repressed angiogenic sprouting/aorta outgrowth. HGF was used as positive control. These results provided direct evidence for PGE$_2$-induced migration as well as angiogenesis of endothelial cell through ROCK II.

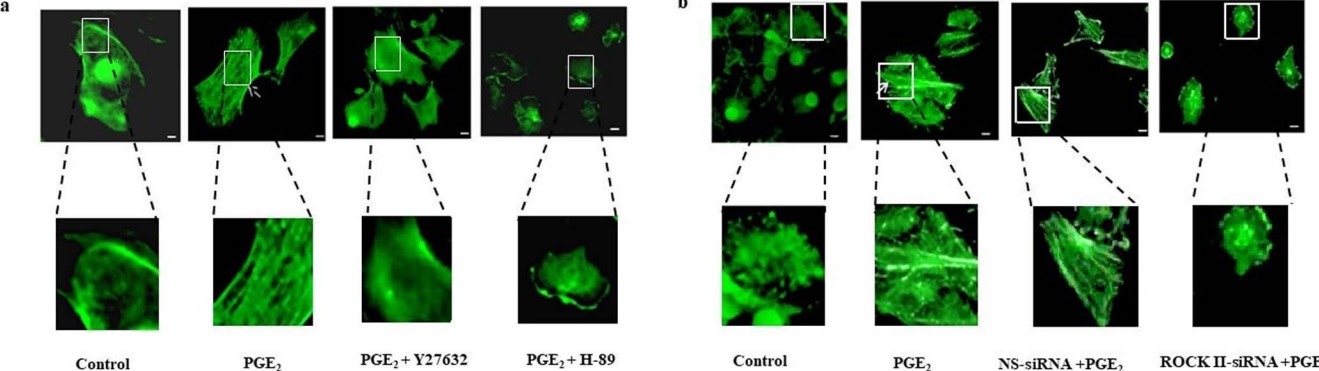

**Fig 6. PGE$_2$-induced actin polymerization and HBECs migration.** HBECs were grown in cover slips and treated with PGE$_2$ in the presence or absence of ROCK II inhibitor (Y27632) and PKA inhibitor (H89). Then incubated with NBD-phallacidin (0.165 μM) for 30 min in dark at room temperature for F-actin staining. Pictures are representative of 5 independent but separate experiments. Magnification of the images in subset is 40X. Images were acquired at 20X and 40X resolution. HBECs were grown in cover slips overnight and transiently transfected with specific siRNA for ROCK II or non-specific (scrambled) RNA for 24 h, followed by treatment with PGE$_2$, fixed with 4% paraformaldehyde and permeabilized with 0.1% Triton X-100, incubated with NBD-phallacidin (0.165 μM) for 30 min in dark at room temperature for F-actin staining. Pictures are representative of 5 independent experiments. Magnification of the images in subset is 40X. Images were acquired at 20X and 40X resolution.

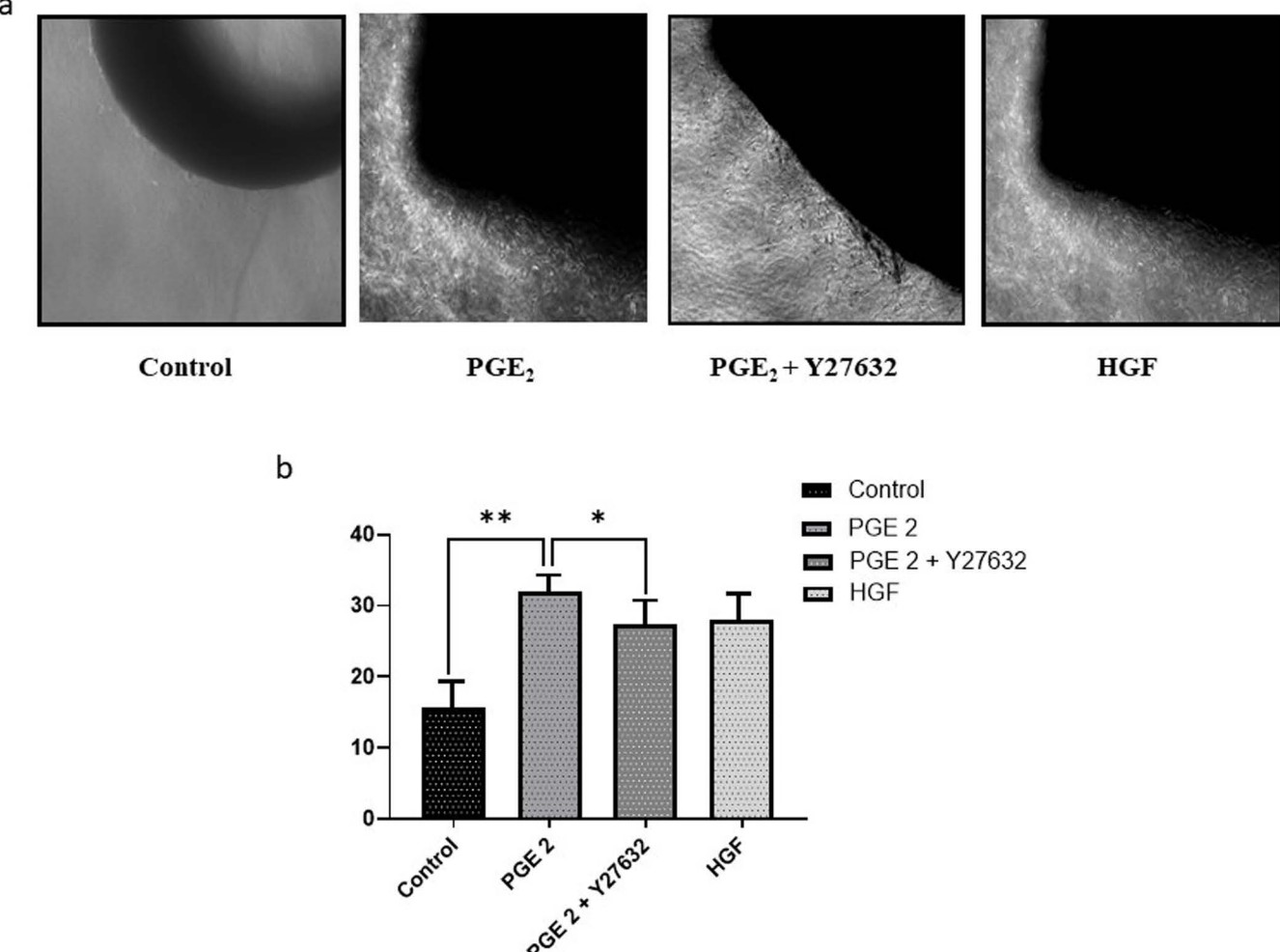

**Fig 7. PGE₂-induced angiogenesis is mediated through ROCK II. (a)** Representative phase contrast images of aortic ring outgrowth after 7 days of treatment with Control (vehicle), PGE₂ (0.1 nM), PGE₂+Y27632 (10 µM, ROCK II inhibitor), or HGF (50 ng/mL, positive control). PGE₂ treatment markedly increased sprouting compared to control, which was reduced upon co-treatment with Y27632. **(b)** Quantification of radial sprouting distance measured using ImageJ. Data are expressed as mean±SEM (n = X rings per group). Statistical analysis was performed using one-way ANOVA followed by Tukey's post hoc test. *p < 0.01, p < 0.05 compared to control.

Quantification of the sprouting response revealed a significant increase in radial outgrowth distance in the PGE₂-treated group compared to control (p < 0.01), confirming the proangiogenic effect of PGE₂. This effect was notably attenuated when aortic rings were pre-treated with the ROCK II inhibitor Y27632, indicating the involvement of the ROCK II signaling pathway (p < 0.05 compared to PGE₂ alone). As expected, treatment with HGF, which was used as a positive control, also promoted substantial sprouting. Fig 7a. These results, consistent with the representative images. Fig 7b is quantitatively summarized, further validating the role of PGE₂ in promoting endothelial sprouting via ROCK II-dependent mechanisms.

## Discussion

Although a large body of evidence supports the critical role of PGE₂ in angiogenesis, few studies have delineated the role of PGE₂ on ECs migration, which is an essential and critical step required for the formation of mature and functional new

blood vessels. In fact, the tuned signaling mechanism that orderly controls the migration of HBECs in response to PGE$_2$ have not yet been fully understood. Previously, we have reported that PGE$_2$ elicits a migratory response of HBECs in a chemotactic manner [7]. We have further demonstrated that, this migration of HBECs by PGE$_2$ is exclusively mediated through its EP$_2$, EP$_3$ and EP$_4$ receptors in a co-operative [7]. Herein, we have demonstrated that PGE$_2$-induced migration of HBECs is regulated through the PKA-ROCK II-MLC pathway.

Our studies have shown that treatment of HBECs with PGE$_2$ induces the expression of ROCK II in a dose-and-time dependent manner. Previous studies had also shown that ROCKs are regulators of cellular apoptosis, metabolism, growth, migration via control of the actin cytoskeletal assembly and cell contraction (Moussavi, Kelley, and Adelstein 1993). It has also been reported that ROCK I mRNA is only expressed in lung, liver, spleen, kidney and testis, whereas ROCK II mRNA is mostly expressed in heart and brain [21,22]. Here, we have also showed that PGE$_2$ induces ROCK II expression. Therefore, the question raised whether the ROCK II has any role in PGE$_2$-induced migration of HBECs. Our pharmacological as well as genetic interference study have shown significant abrogation of PGE$_2$-induced migration of HBECs. Hence, it is the first kind of a study where we have shown that PGE$_2$-dependent migration of HBECs is mediated through ROCK II.

Previously, we have also shown that PGE$_2$-induced migration of HBECs is mainly mediated through EP receptors-cAMP-PKA signaling axis in a co-operative manner [7]. Therefore, attempts have been made to explore whether the cAMP-PKA signaling activates ROCK II, followed by MLC phosphorylation and migration. It is known that phosphorylation of MLC plays a critical role in controlling actinomyosin contractility [23–25]. The equilibrium of MLC phosphorylation is also tightly controlled by myosin light chain kinase and myosin phosphatase [26]. A study has showed that ROCK II plays a dual role in the contraction of muscle cells through MLC phosphorylation, and at the same time inhibiting the myosin phosphatase [27]. Our data showed that PGE$_2$ induces MLC phosphorylation in a dose-and-time dependent manner. We have also shown that pharmacological inhibition of ROCK II (by ROCK II specific inhibitor Y27632) or siRNA mediated ROCK II silencing, significantly abrogated PGE$_2$-induced MLC phosphorylation. Therefore, this study confirms that PGE$_2$-induced migration of HBECs is mediated through cAMP-PKA which then triggers ROCK II activation and consequently leads to MLC phosphorylation.

The three major cytoskeletal systems found in all animal cells, which include the actin cytoskeleton, microtubule network, and intermediate filaments, appear to coordinate their functions in mediating numerous cellular processes [28]. This coordination or crosstalk is achieved through a number of mechanisms, such as Rac1 and Rho GTPases, their regulators including GEFs and GAPs, or their downstream effectors, associated with both the actin cytoskeleton and microtubules in a competitive manner [29,30]. Cellular migration is a dynamic complex process in adherent cells which can be viewed as a cycle of extension, attachment and detachment. In order to facilitate cellular migration, ECs generate required friction and motive force [31]. However, the role of PGE$_2$ in actin polymerization/rearrangement through ROCK II is still unknown. Our pharmacological inhibition study of ROCK II (by ROCK II specific inhibitor Y27632) or PKA (H-89; PKA inhibitor) or siRNA mediated ROCK II silencing, significantly abrogated PGE$_2$-induced actin polymerization/ rearrangement when compared to the control.

Migration of ECs plays a pivotal role in the reconstruction of new blood vessels through the arrangement of ECs lining which could be observed in *ex-vivo* aortic ring assay [32]. This prompted us to study the role of PGE$_2$-induced new blood vessels formation through ROCK II, and thus the *ex-vivo* aortic ring angiogenesis assay was performed in the presence of ROCK II inhibitor (Y27632). Our study has shown that pre-treatment of ROCK II inhibitor significantly inhibits PGE$_2$-induced ECs spouting. HGF was used as a positive control.

In conclusion, the results of this investigation are consistent with the hypothesis that PGE$_2$-induced migration of ECs and new blood vessels formation is mediated through cAMP-PKA-ROCK II-MLC phosphorylation and F-actin polymerization. Therefore, this study provides strong support for the development of targeted strategies to treat cerebrovascular diseases that are associated with dysregulated angiogenesis.

## Acknowledgments

The authors would like to express their gratitude to King Faisal University for its support in providing conducive research and a learning environment, which facilitated the successful completion of this research work. This work was supported by the Deanship of Scientific Research, Vice Presidency for Graduate Studies and Scientific Research, King Faisal University, Saudi Arabia [Grant No. KFU 252130 to GAK]

## Author contributions

**Conceptualization:** Gausal Azam Khan.

**Data curation:** Gausal Azam Khan.

**Formal analysis:** Gausal Azam Khan, Arjun Ghosh.

**Funding acquisition:** Gausal Azam Khan.

**Investigation:** Gausal Azam Khan.

**Methodology:** Gausal Azam Khan, Arjun Ghosh.

**Project administration:** Gausal Azam Khan.

**Software:** Arjun Ghosh.

**Supervision:** Gausal Azam Khan.

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
