## [Decision Letter · Decision Letter 0]

PONE-D-25-14301Prostaglandin E2 dependent migration of Human Brain Endothelial Cells is mediated through Rho-Kinase-IIPLOS ONE

Dear Dr. Khan,

Thank you for submitting your manuscript to PLOS ONE. After careful consideration, we feel that it has merit but does not fully meet PLOS ONE’s publication criteria as it currently stands. Therefore, we invite you to submit a revised version of the manuscript that addresses the points raised during the review process.

We look forward to receiving your revised manuscript.

Kind regards,

Yoshi

Prof. Yoshinori Marunaka, MD. PhD

Academic Editor

PLOS ONE

https://academic.oup.com/jleukbio/article-abstract/105/4/705/6935763?redirectedFrom=fulltext&login=false

https://d.docksci.com/post-polymerization-crosstalk-between-the-actin-cytoskeleton-and-microtubule-net_5a0eb427d64ab2eca1fa284b.html

https://www.thno.org/v08p6053.htm

In your revision ensure you cite all your sources (including your own works), and quote or rephrase any duplicated text outside the methods section. Further consideration is dependent on these concerns being addressed.

3. To comply with PLOS ONE submissions requirements, in your Methods section, please provide additional information regarding the experiments involving animals and ensure you have included details on (1) methods of sacrifice, and (2) efforts to alleviate suffering.

4. Please include a complete copy of PLOS’ questionnaire on inclusivity in global research in your revised manuscript. Our policy for research in this area aims to improve transparency in the reporting of research performed outside of researchers’ own country or community. The policy applies to researchers who have travelled to a different country to conduct research, research with Indigenous populations or their lands, and research on cultural artefacts. The questionnaire can also be requested at the journal’s discretion for any other submissions, even if these conditions are not met.  Please find more information on the policy and a link to download a blank copy of the questionnaire here: https://journals.plos.org/plosone/s/best-practices-in-research-reporting. Please upload a completed version of your questionnaire as Supporting Information when you resubmit your manuscript.

“This work was supported by grants DSR KFU”

6. We note that your Data Availability Statement is currently as follows: [All relevant data are within the manuscript and its supporting information files.]

7. PLOS ONE now requires that authors provide the original uncropped and unadjusted images underlying all blot or gel results reported in a submission’s figures or Supporting Information files. This policy and the journal’s other requirements for blot/gel reporting and figure preparation are described in detail at https://journals.plos.org/plosone/s/figures#loc-blot-and-gel-reporting-requirements and https://journals.plos.org/plosone/s/figures#loc-preparing-figures-from-image-files. When you submit your revised manuscript, please ensure that your figures adhere fully to these guidelines and provide the original underlying images for all blot or gel data reported in your submission. See the following link for instructions on providing the original image data: https://journals.plos.org/plosone/s/figures#loc-original-images-for-blots-and-gels.  

Reviewers' comments:

Reviewer's Responses to Questions

**Comments to the Author**

1. Is the manuscript technically sound, and do the data support the conclusions?

Reviewer #1: Yes

Reviewer #2: Yes

2. Has the statistical analysis been performed appropriately and rigorously? 

Reviewer #1: Yes

Reviewer #2: Yes

3. Have the authors made all data underlying the findings in their manuscript fully available?

Reviewer #1: Yes

Reviewer #2: Yes

4. Is the manuscript presented in an intelligible fashion and written in standard English?

Reviewer #1: Yes

Reviewer #2: Yes

5. Review Comments to the Author

Reviewer #1: It is well structured and nicely written research work. Similar work has been done before but you have added animal study to it. The work is generally acceptable but you need to make it concise and to the point with results and findings more magnified.

Reviewer #2: In the current manuscript, the authors report that Prostaglandin E2 (PGE2) plays a crucial role in the migration of Human Brain Endothelial Cells through the activation of Rho Kinase II (ROCK-II). The paper is well written and organized. However, following are the concerns:

Major comments

1. In Figure 5a, the pMLC band after the addition of PGE2 appears thinner and weaker than the bands observed under the same conditions in Figures 5b and 5c (+PGE2 lanes). It is unclear why such a discrepancy occurs. Additionally, the authors state that the membrane used to detect pMLC was reprobed to detect total MLC. However, the shapes and widths of the respective bands appear to differ (for example, in Figure 5c). If the same membrane was used for both detections, the band shapes for pMLC and MLC should be identical.

If different membranes were used instead, the authors should clearly indicate this and ideally provide results obtained from the same membrane.

2. In Figure 6, the image showing actin polymerization is not very clear.

In particular, in Figure 6b, the authors state that “NS-siRNA has no effect,” but the development of F-actin is not readily discernible.

If a higher-quality image is available, I would suggest replacing the current one.

3. The authors performed a mouse aortic ring sprouting assay and showed that pretreatment with Y27632 repressed angiogenic sprouting/aorta outgrowth in Figure 7.

However, I was unable to understand how they reached this conclusion.

Would it be possible to quantitatively analyze the results as done in the following paper?

https://doi.org/10.1016/j.mbplus.2020.100025

Minor comments

1. The order of the samples in the PCR results is confusing (in Figure 1). Could this be corrected?

2. I would like to see the units standardized. For example, “0.1 nM” in the figure and “100 pM” in the text (page 10, line 296 and 300), etc.

3. There is no Figure 3c, but the text refers to Fig 3c in page 11 (is it a mistake for Fig 3b?).

6. PLOS authors have the option to publish the peer review history of their article (what does this mean? ). If published, this will include your full peer review and any attached files.

**Do you want your identity to be public for this peer review?** For information about this choice, including consent withdrawal, please see our Privacy Policy .

Reviewer #1: **Yes**

Reviewer #2: No

---

## [Author Response · Author response to Decision Letter 1]

13 May 2025

Reviewer’s comments

Reviewer #1:

Comment:

It is well structured and nicely written research work. Similar work has been done before but you have added animal study to it. The work is generally acceptable but you need to make it concise and to the point with results and findings more magnified.

Response:

Thank you for your valuable comments and constructive suggestions on our manuscript. We appreciate your positive feedback. In response, we have revised the manuscript to improve its clarity, conciseness, and overall focus. The edited sections have been clearly highlighted for your convenience. We have also refined the presentation of our results and findings to ensure better clarity and scientific rigor. Furthermore, we have enhanced the quality of several figures to improve readability and facilitate a clearer understanding of the data presented.

Reviewer #2:

In the current manuscript, the authors report that Prostaglandin E2 (PGE2) plays a crucial role in the migration of Human Brain Endothelial Cells through the activation of Rho Kinase II (ROCK-II). The paper is well written and organized. However, following are the concerns:

Comment 1:

In Figure 5a, the pMLC band after the addition of PGE2 appears thinner and weaker than the bands observed under the same conditions in Figures 5b and 5c (+PGE2 lanes). It is unclear why such a discrepancy occurs. Additionally, the author’s state that the membrane used to detect pMLC was reprobed to detect total MLC. However, the shapes and widths of the respective bands appear to differ (for example, in Figure 5c). If the same membrane was used for both detections, the band shapes for pMLC and MLC should be identical. If different membranes were used instead, the authors should clearly indicate this and ideally provide results obtained from the same membrane.2.

Response 1:

We thank the reviewer comments/concern. We have modified Figure 5a accordingly. Separate experiments were conducted for pMLC and total PMC blotting, and each was reprobed with β-Actin antibody independently as a loading control. This clarification has been incorporated into the Materials and Methods section under the subheading Immunoblotting (page 8, lines 236–242), as well as in the figure legend page (page 22, lines 726–731) in the revised manuscript.

Comments 2:

In Figure 6, the image showing actin polymerization is not very clear. In particular, in Figure 6b, the authors state that “NS-siRNA has no effect,” but the development of F-actin is not readily discernible. If a higher-quality image is available, I would suggest replacing the current one.

Response 2:

We thank the reviewer for their valuable observation regarding Figure 6a & 6b. To improve the clarity and visibility of F-actin polymerization, we have updated the figures with higher-magnification images in BOX. Specifically, images were originally acquired at both 20X and 40X magnification. In the revised version, we selected a representative region from the 20X image and magnified it using 40X to enhance resolution and provide a clearer visualization of F-actin structures. This adjustment should allow for better assessment of the statement that "NS-siRNA has no effect." We hope the revised images address the concern effectively. (page 22, line 750-751 and page 23 line 756-757)

Comments 3:

The authors performed a mouse aortic ring sprouting assay and showed that pre-treatment with Y27632 repressed angiogenic sprouting/aorta outgrowth in Figure 7.However, I was unable to understand how they reached this conclusion. Would it be possible to quantitatively analyse the results as done in the following paper? https://doi.org/10.1016/j.mbplus.2020.100025

Response 3:

We thank you the reviewer for the comments. The mouse aortic ring assay is a well-established ex vivo model that recapitulates key processes of angiogenesis, including endothelial cell proliferation, migration, and microvessel sprouting. In our study, we utilized this assay to assess angiogenic activity, employing quantification methods aligned with validated protocols from prior literature. For instance, Nunes et al. (2020) [DOI: 10.1016/j.mbplus.2020.100025] demonstrated that microvessel outgrowth from aortic rings reliably reflects angiogenic potential, allowing for robust quantitative analysis in response to pro- and anti-angiogenic stimuli. Similarly, the PLOS ONE study https://journals.plos.org/plosone/article/figure?id=10.1371/journal.pone.0069552.g002] showed that metrics such as sprout number and cumulative sprout length are sensitive and reliable indicators of angiogenic response. Consistent with these studies, we quantified microvessel sprouting using analogous parameters, including the number and total length of sprouts. This methodological consistency reinforces the validity of our findings and supports the conclusion that the observed sprouting reflects angiogenic induction. Accordingly, we have add and discussed (page 9, line 274-284.Page 14, line 411-419) and Figure 7b legend (page 23, line 767-770) & Two references were added (page 18, line 566-569 & page 20 line 633-636)

Minor comments:

1. The order of the samples in the PCR results is confusing (in Figure 1). Could this be corrected?

Response 1:

Thank you for your valuable comments. We have carefully addressed your feedback and have updated Figure 1 to reflect the correct markings and sequence. Additionally, we have included a box around the relevant area to enhance clarity and ensure the sequence is easily identifiable.

We appreciate your attention to detail and believe these revisions improve the overall presentation of the data.

2. I would like to see the units standardized. For example, “0.1 nM” in the figure and “100 pM” in the text (page 10, line 296 and 300), etc.

Response 2:

Thank you for taking the time to review our work and for providing detailed and insightful comments. We have addressed the unit discrepancy between the Results section and the figure by standardizing all relevant units to "nM." These revisions have been implemented in the updated manuscript, specifically (page 11, lines 317 and 321).

3. There is no Figure 3c, but the text refers to Fig 3c in page 11 (is it a mistake for Fig 3b?).

Response 3:

Thank you for your careful review of our manuscript and for pointing out the error in Figure 3. We appreciate your attention to detail. You are correct that it was incorrectly labelled as 3c instead of 3b. This mistake has been rectified in the revised manuscript, as reflected (page 12, line 343-344).

---

## [Decision Letter · Decision Letter 1]

PONE-D-25-14301R1Prostaglandin E2 dependent migration of Human Brain Endothelial Cells is mediated through Rho-Kinase-IIPLOS ONE

Dear Dr. Khan,

Thank you for submitting your manuscript to PLOS ONE. After careful consideration, we feel that it has merit but does not fully meet PLOS ONE’s publication criteria as it currently stands. Therefore, we invite you to submit a revised version of the manuscript that addresses the points raised during the review process.

We look forward to receiving your revised manuscript.

Kind regards,

Yoshi

Prof. Yoshinori Marunaka, MD. PhD

Academic Editor

PLOS ONE

Journal Requirements:

Reviewers' comments:

Reviewer's Responses to Questions

**Comments to the Author**

1. If the authors have adequately addressed your comments raised in a previous round of review and you feel that this manuscript is now acceptable for publication, you may indicate that here to bypass the “Comments to the Author” section, enter your conflict of interest statement in the “Confidential to Editor” section, and submit your "Accept" recommendation.

Reviewer #1: All comments have been addressed

Reviewer #2: All comments have been addressed

2. Is the manuscript technically sound, and do the data support the conclusions?

Reviewer #1: Yes

Reviewer #2: Yes

3. Has the statistical analysis been performed appropriately and rigorously? 

Reviewer #1: Yes

Reviewer #2: Yes

4. Have the authors made all data underlying the findings in their manuscript fully available?

Reviewer #1: Yes

Reviewer #2: Yes

5. Is the manuscript presented in an intelligible fashion and written in standard English?

Reviewer #1: Yes

Reviewer #2: Yes

6. Review Comments to the Author

Reviewer #1: (No Response)

Reviewer #2: The higher magnification image in Figure 6b is incorrect.

The high magnification image of “NS-siRNA + PGE2” is identical to that of PGE2 + Y27632 in Figure 6a.

Please check the image carefully again.

7. PLOS authors have the option to publish the peer review history of their article (what does this mean? ). If published, this will include your full peer review and any attached files.

**Do you want your identity to be public for this peer review?** For information about this choice, including consent withdrawal, please see our Privacy Policy .

Reviewer #1: **Yes: ** amjed abbawe salih

Reviewer #2: No

---

## [Author Response · Author response to Decision Letter 2]

26 May 2025

Reviewer’s comments

Comments:

Please review your reference list to ensure that it is complete and correct. If you have cited papers that have been retracted, please include the rationale for doing so in the manuscript text, or remove these references and replace them with relevant current references. Any changes to the reference list should be mentioned in the rebuttal letter that accompanies your revised manuscript. If you need to cite a retracted article, indicate the article has retracted status in the References list and include a citation and full reference for the retraction notice.

Response:

We sincerely thank the editor for the valuable comments and for highlighting important concerns regarding the references. In response, we have conducted a thorough review of all references cited in the manuscript. We confirm that none of the cited articles have been retracted. All references are recent, relevant, and appropriately aligned with the context, rationale, and scientific content of the manuscript. Additionally, we have ensured consistency and accuracy in formatting using Mendeley Reference Manager, adhering to the PLOS ONE citation style. The revised and verified reference list is provided on pages 17–19, lines 532–620.

Reviewer #2:

Comment:

The higher magnification image in Figure 6b is incorrect. The high magnification image of “NS-siRNA + PGE2” is identical to that of PGE2 + Y27632 in Figure 6a.

Please check the image carefully again.

Response:

We thank the reviewer for their valuable observation regarding Figures 6a and 6b. In response, we have carefully revised and corrected the figures. The updated versions accurately reflect the intended data, and no concerns remain regarding their content.

---

## [Decision Letter · Decision Letter 2]

Prostaglandin E2 dependent migration of Human Brain Endothelial Cells is mediated through Rho-Kinase-II

PONE-D-25-14301R2

Dear Prof. Gausal Azam Khan,

We’re pleased to inform you that your manuscript has been judged scientifically suitable for publication and will be formally accepted for publication once it meets all outstanding technical requirements.

Kind regards,

Yoshi

Prof. Yoshinori Marunaka, MD. PhD

Academic Editor

PLOS ONE

Additional Editor Comments (optional):

Reviewers' comments:

Reviewer's Responses to Questions

**Comments to the Author**

1. If the authors have adequately addressed your comments raised in a previous round of review and you feel that this manuscript is now acceptable for publication, you may indicate that here to bypass the “Comments to the Author” section, enter your conflict of interest statement in the “Confidential to Editor” section, and submit your "Accept" recommendation.

Reviewer #2: All comments have been addressed

2. Is the manuscript technically sound, and do the data support the conclusions?

Reviewer #2: Yes

3. Has the statistical analysis been performed appropriately and rigorously? 

Reviewer #2: Yes

4. Have the authors made all data underlying the findings in their manuscript fully available?

Reviewer #2: Yes

5. Is the manuscript presented in an intelligible fashion and written in standard English?

Reviewer #2: Yes

6. Review Comments to the Author

Reviewer #2: (No Response)

7. PLOS authors have the option to publish the peer review history of their article (what does this mean? ). If published, this will include your full peer review and any attached files.

**Do you want your identity to be public for this peer review?** For information about this choice, including consent withdrawal, please see our Privacy Policy .

Reviewer #2: No

---

## [Editor Report · Acceptance letter]

PONE-D-25-14301R2

PLOS ONE

Dear Dr. Khan,

I'm pleased to inform you that your manuscript has been deemed suitable for publication in PLOS ONE. Congratulations! Your manuscript is now being handed over to our production team.

Kind regards,

on behalf of

Professor Yoshinori Marunaka

Academic Editor

PLOS ONE